# On the Kaniadakis Distributions Applied in Statistical Physics and Natural Sciences

**DOI:** 10.3390/e25020292

**Published:** 2023-02-04

**Authors:** Tatsuaki Wada, Antonio Maria Scarfone

**Affiliations:** 1Region of Electrical and Electronic Systems Engineering, Ibaraki University, Nakanarusawa-cho, Hitachi-shi 316-8511, Japan; 2Istituto dei Sistemi Complessi, Consiglio Nazionale delle Ricerche (ISC-CNR), c/o Politecnico di Torino, Corso Duca degli Abruzzi, 24, 10129 Torino, Italy

**Keywords:** κ-deformed functions, constitutive relations, Gompertz rule, Lotka–Volterra equations, contact density dynamics

## Abstract

Constitutive relations are fundamental and essential to characterize physical systems. By utilizing the κ-deformed functions, some constitutive relations are generalized. We here show some applications of the Kaniadakis distributions, based on the inverse hyperbolic sine function, to some topics belonging to the realm of statistical physics and natural science.

## 1. Introduction

The κ-exponential function [1,2,3] is defined by:(1)$$\begin{array}{c} \exp_{\kappa }\left(x\right):={\left(\kappa x+\sqrt{1+\kappa^{2}x^{2}}\right)}^{\frac{1}{\kappa }}=\exp\left[\frac{1}{\kappa }\mathrm{arsinh}\left(\kappa x\right)\right], \end{array}$$
for a real deformation parameter κ. The inverse function, i.e., the κ-deformed logarithmic function, is defined by:(2)$$\begin{array}{c} \ln_{\kappa }x:=\frac{x^{\kappa }-x^{-\kappa }}{2\kappa }=\frac{1}{\kappa }\sinh\left[\kappa \ln x\right]. \end{array}$$

Both κ-deformed functions are important ingredients of the generalized statistical physics based on κ-entropy [1,2,3]. This influences a wide range of scientific fields, and, based on the κ-deformed functions (Appendix A), several basic fields developed over two decades. Kaniadakis [4] provided the theoretical foundations and mathematical formalism generated by the κ-deformed functions, and some references, including many fields of applications. Recently, the usefulness of the κ-statistics was demonstrated for the analysis [5] of epidemics and pandemics.

Constitutive relations are fundamental and essential to characterize physical systems. They are combined with the other equations of the physical laws in order to solve physical problems. There are well-known examples of linear constitutive relations, such as the following: Hooke’s law $F=k_{s}x$, for the tensile, or compressive, force *F* of a spring with a spring constant $k_{s}$ against the change in its length *x*; Ohm’s law V=RI for the voltage *V* of an electrical conductor with resistance *R* under an electric current *I*, and so on. However, as a real spring deviates from Hooke’s law, we know that any linear constitutive relation describes an idealized situation, and it is merely a linearized- and/or approximated- relation to describe some real physical properties. Hence, in general, non-linearity plays a crucial role to describe more realistic physical systems.

The κ-exponential function (Equation 1) can be regarded as a useful tool (or device) to make such non-linear constitutive relations for a better description of real physical systems. For example, consider the following κ-deformation of Hooke’s law:(3)$$\begin{array}{c} F_{\kappa }:=k_{s}\ln\left[\exp_{\kappa }\left(x\right)\right]=\frac{k_{s}}{\kappa }\ln\left(\kappa x+\sqrt{1+\kappa^{2}x^{2}}\right), \end{array}$$
which reduces to the original Hooke’s law $F=k_{s}x$ in the limit of κ→0. For any linear constitutive relation, we can apply this type of the κ-deformation. For example, Ohm’s law can be cast into the following form: $V=RI=R\ln\left[\exp(I)\right]$. By changing the exponential function with the κ-exponential function, we obtain the κ-deformed version of Ohm’s law: $V_{\kappa }=R\ln\left[\exp_{\kappa }\left(I\right)\right]$. In this research, we focused on this type of the κ-deformation of a physical quantity (say *A*), i.e.,
(4)$$\begin{array}{c} A\;\Rightarrow \;\ln\left[\exp_{\kappa }\left(A\right)\right]=\frac{1}{\kappa }\mathrm{arsinh}\left(\kappa A\right). \end{array}$$

Throughout this paper, we call this κ-deformation *the arsinh-type deformation* of a physical quantity *A*.

Another type of the κ-deformation can be:(5)$$\begin{array}{c} A\;\Rightarrow \;\ln_{\kappa }\left[\exp(A)\right]=\frac{1}{\kappa }\sinh\left(\kappa A\right), \end{array}$$
which is called here *the sinh-type deformation*. In Reference [6], the thermodynamic stability of the κ-generalization $S_{\kappa }^{B}$ of Boltzmann entropy $S^{B}$ was studied. The κ-generalization $S_{\kappa }^{B}$ was rewritten in the form:(6)$$\begin{array}{c} S_{\kappa }^{B}:=k_{B}\ln_{\kappa }W=k_{B}\ln_{\kappa }\left[\exp\left(\ln W\right)\right]=k_{B}\ln_{\kappa }\left[\exp\left(S^{B}\right)\right], \end{array}$$
which could be regarded as the sinh-type deformation of Boltzmann entropy $S^{B}$. Recently, in cosmology, Lymperis et al. [7] modified Bekenstein–Hawking entropy $S^{\mathrm{BH}}$ as follows:(7)$$\begin{array}{c} S_{\kappa }^{\mathrm{BH}}=\frac{1}{\kappa }\sinh\left(\kappa S^{\mathrm{BH}}\right), \end{array}$$
which was obviously the sinh-type deformation of $S^{\mathrm{BH}}$.

In this paper we considered the arsinh-type deformations against some constitutive relations in the field of statistical physics and natural sciences. In our previous work [8] we studied a thermal particle under a velocity-dependent potential which could be regarded as a deformation of Rayleigh’s dissipation function [9] and showed that the probability distribution function (pdf) for the stationary-state of this thermal particle was a κ-deformed Gaussian pdf. It was considered the canonical pdf ρ(v), in the velocity space, of a thermal particle with unit mass (m=1) in the κ-deformed confining potential $U_{\kappa \beta }\left(v\right)$:(8)$$\begin{array}{cc} U_{\kappa \beta }\left(v\right) & :=\frac{1}{\kappa \beta }\mathrm{arsinh}\left(\kappa \beta \frac{v^{2}}{2}\right), \end{array}$$
where $\beta :=1/k_{B}T$ is a coldness (or inverse temperature). This κ-deformed potential $U_{\kappa \beta }\left(v\right)$ was rewritten, in the momentum–space, as:(9)$$\begin{array}{c} U_{\kappa \beta }\left(p\right)=\frac{1}{\kappa \beta }\mathrm{arsinh}\left(\kappa \beta \frac{p^{2}}{2}\right)=\frac{1}{\beta }\ln\left[\exp_{\kappa }\left(\beta \frac{p^{2}}{2}\right)\right], \end{array}$$
which was the arsinh type deformation of the quantity $\beta p^{2}/2$ (the ratio of the kinetic energy to the mean thermal energy $k_{B}T=1/\beta$). In other words, we considered the following κ-deformation $Q_{\kappa }\left(U\right)$ of the Boltzmann factor exp(−βU) for an equilibrium state with the energy *U*:(10)$$\begin{array}{c} Q_{\kappa }\left(U\right):=\exp_{\kappa }\left(-\beta U\right)=\exp\left[\frac{1}{\kappa }\mathrm{arsinh}\left(-\kappa \beta U\right)\right]. \end{array}$$

One may wonder why the inverse hyperbolic sine function (arsinh) plays a role. In many different fields of sciences, there is no doubt that the exponential and logarithmic functions are important and fundamental. Since the inverse hyperbolic sine function and logarithmic function are mutually related as:(11)$$\begin{array}{c} \mathrm{arsinh}x=\ln\left[x+\sqrt{1+x^{2}}\right],\;\ln x=\mathrm{arsinh}\left[\frac{1}{2}\left(x-\frac{1}{x}\right)\right], \end{array}$$
for a positive real *x*, we think both functions are important. By using the second relation, for any real parameter κ≠0, we have:(12)$$\begin{array}{c} \ln x=\frac{1}{\kappa }\ln x^{\kappa }=\frac{1}{\kappa }\mathrm{arsinh}\left[\frac{1}{2}\left(x^{\kappa }-x^{-\kappa }\right)\right]=\frac{1}{\kappa }\mathrm{arsinh}\left[\kappa \ln_{\kappa }x\right]. \end{array}$$

Note that this relation corresponds to the arsinh-type deformation of $\ln_{\kappa }x$ and is equivalent to definition (Equation 2) of the κ-deformed logarithmic function that can be regarded as the sinh-type of κ-deformation of lnx. Kaniadakis already discussed this issue in section II of Reference [2] from the viewpoint of deformed algebra.

On the other hand, Pistone [10] was the first one to study the κ-exponential model in the field of information geometry [11], and later, through our research activities [8,12,13], we realized that there exist some relations among statistical physics, thermodynamics, mathematical biology, and information geometry. Harper [14,15] pointed out that the replicator equation (RE) [16] in mathematical biology or in an evolutional game theory [17] is related with information geometry and a general form of the Lotka–Volterra (gLV) equation as briefly explained in Appendix B. The gLV equations [14,15,18,19]:(13)$$\begin{array}{c} \frac{dy_{i}}{dt}=y_{i}\;f_{i}\left(y\right), \end{array}$$
are used to model the competition dynamics of the populations $y_{1},y_{2},\ldots ,y_{n}$ of *n* biological species. The Gompertz function [20] is a type of mathematical model for time evolution. Historically, he studied human mortality and proposed his law of human mortality in which he assumed that a person’s resistance to death decreases as his or her years increase. His law is now called *Gompertz rule* (or law) and we would like to point out the relation of his function and his rule to some important quantities concerning statistical physics.

The rest of the paper is organized as follows. In Section 2, we briefly explain Gompertz function, and the gLV equations, which are important in mathematical biology (or evolutional game theory). Their relations to thermal physics are pointed out. Section 3 considers the thermal density operator, which is characterized by the so-called Bloch equation [21,22] for thermal states, and we show that the Bloch equation can be regarded as a Gompertz rule after the parameter transformation β to t=−lnβ. In Section 4, we discuss the arsinh-type deformation from the viewpoint of the κ-addition. In Section 5, we study the numerical simulations of the thermostat algorithm for the Hamiltonian with the κ-deformed kinetic energy, which can be regarded as the arsinh type of the κ-deformation of the ratio $\beta p^{2}/2$ as shown in (Equation 10). The final section is devoted to our conclusions.

## 2. Gompertz Functions and Gompertz Rule

Here we would like to point out that there exist relations between evolutional game dynamics and thermal physics. In evolutional game theory [17], evolutional game dynamics is described by a RE. The gLV equations are related to REs, as shown in Appendix B. On the other hand, Gompertz function is a mathematical model describing an evolutional curve. Gompertz function (or Gompertz curve) [20] is a type of mathematical model for a time series. Gompertz function $f_{G}\left(t\right)$ is a sigmoid function and is given by:(14)$$\begin{array}{c} f_{G}\left(t\right):=K\exp\left[C\;\exp(-t)\right], \end{array}$$
where *C* and *K* are positive constants. A distinctive feature of Gompertz function is its double exponential *t*-dependency. His function is nowadays used in many different areas to model time evolution of populations where growth is slowest at the start and end of a period. For example, Reference [23] applied Gompertz model to describe the growth dynamics of the COVID-19 pandemic. Gompertz [20] studied human mortality by working out a series of mortality tables, and this suggested to him his law of human mortality, in which he assumed that a person’s resistance to death decreases as age increases. The rule of his model is called *Gompertz rule* which states that:(15)$$\begin{array}{c} \frac{d}{dt}f_{G}\left(t\right)=-f_{G}\left(t\right)\ln\frac{f_{G}\left(t\right)}{K}. \end{array}$$

The solution of the Gompertz rule is the Gompertz function (Equation 14), if we set $K=\lim_{t\to \infty }f_{G}\left(t\right)$ and $C=\ln(f_{G}\left(0\right)/K)$.

If we choose $f_{i}\left(y\left(t\right)\right)=-\ln y_{i}\left(t\right)$ and assume $\lim_{t\to \infty }y_{i}\left(t\right)=1$, the gLV Equation (Equation 13) becomes:(16)$$\begin{array}{c} \frac{dy_{i}\left(t\right)}{dt}=-y_{i}\left(t\right)\;\ln y_{i}\left(t\right), \end{array}$$
which can be regarded as the Gompertz rule (Equation 15) with K=1 for each $y_{i}\left(t\right)$. Consequently, its solution $y_{i}\left(t\right)$ is the Gompertz function:(17)$$\begin{array}{c} y_{i}\left(t\right)=\exp\left[\ln y_{i}\left(0\right)\;\exp\left(-t\right)\right]. \end{array}$$

Now, by changing the parameter *t* to β=exp(−t), we have dβ=−βdt so that the limit t→0 corresponds to β→1, and each constant $E_{i}$ is introduced as:(18)$$\begin{array}{c} -E_{i}=\lim_{t\to 0}\ln y_{i}\left(t\right)=\lim_{\beta \to 1}\ln y_{i}\left(\beta \right), \end{array}$$
where $y_{i}\left(\beta \right)$ is the shorthand notation of $y_{i}\left(t\left(\beta \right)\right)$ with t(β)=−lnβ. Then, the solution $y_{i}\left(\beta \right)$ in (Equation 17) can be expressed as a quantity very familiar to statistical physics:(19)$$\begin{array}{c} y_{i}\left(\beta \right)=\exp\left(-\beta E_{i}\right), \end{array}$$
that is the Boltzmann factor. The corresponding Gompertz rule (Equation 15) for $y_{i}\left(\beta \right)$ is equivalent to:(20)$$\begin{array}{c} \frac{d}{d\beta }y_{i}\left(\beta \right)=-E_{i}\;y_{i}\left(\beta \right). \end{array}$$

Having described the relation between the Gompertz rule and the Boltzmann factor $\exp(-\beta E_{i})$ in statistical physics, in the next section we discuss a κ-deformation of the Bloch equation for thermal states.

## 3. Bloch Equation for Thermal States

For a given Hamiltonian $\hat{H}$ and the corresponding eigenvalues $E_{i}$ and eigenstate $|\psi_{i}\rangle$, which are related in:(21)$$\begin{array}{c} \hat{H}|\psi_{i}\rangle =E_{i}|\psi_{i}\rangle , \end{array}$$
and assuming the completeness relation $\sum_{i}|\psi_{i}\rangle \langle \psi_{i}|=\hat{1}$, the density operator $\hat{\rho }\left(\beta \right)$ for a canonical ensemble is constructed as:(22)$$\begin{array}{c} \hat{\rho }\left(\beta \right):=\sum_{i}\exp\left(-\beta E_{i}\right)|\psi_{i}\rangle \langle \psi_{i}|=\exp\left(-\beta \hat{H}\right). \end{array}$$
In order to determine the canonical density matrix, we have to solve the eigenvalue Equation (Equation 21) and to sum over all the states. This needs heavy calculations in general. Note that $\hat{\rho }\left(\beta \right)$ is un-normalized and its trace is $\mathrm{Tr}\hat{\rho }\left(\beta \right)=Z\left(\beta \right)$, which is the partition function.

The Bloch equation [21,22] for thermal states is known as:(23)$$\begin{array}{c} -\frac{\partial }{\partial \beta }\hat{\rho }\left(\beta \right)=\hat{H}\;\hat{\rho }\left(\beta \right), \end{array}$$
which can be regarded as the diffusion equation in imaginary time β, and it has a similar form as Schrödinger equation and diffusion equation. Bloch Equation (Equation 23) offers an alternative route to determine the density operator $\hat{\rho }\left(\beta \right)$. The initial (β=0) condition is provided if we know the eigenstates in the high-temperature limit.

Now, by multiplying β to both sides of (Equation 23), we have:(24)$$\begin{array}{c} -\beta \frac{\partial }{\partial \beta }\hat{\rho }\left(\beta \right)=\beta \hat{H}\;\hat{\rho }\left(\beta \right)=-\ln\left[\hat{\rho }\left(\beta \right)\right]\;\hat{\rho }\left(\beta \right). \end{array}$$
Changing the parameter β to t=−lnβ, it follows:(25)$$\begin{array}{c} \frac{d}{dt}\hat{\rho }\left(t\right)=-\beta \frac{d}{d\beta }\hat{\rho }\left(\beta \right)=-\ln\left[\hat{\rho }\left(t\right)\right]\;\hat{\rho }\left(t\right). \end{array}$$
This is the same form of the Gompertz rule (Equation 15). In this way, the Bloch equation can be considered as a sort of Gompertz rule.

Next, let us consider the κ-deformed density operator:(26)$$\begin{array}{c} {\hat{\rho }}_{\kappa }\left(\beta \right):=\sum_{i}\exp_{\kappa }\left(-\beta E_{i}\right)|\psi_{i}\rangle \langle \psi_{i}|=\exp_{\kappa }\left(-\beta \hat{H}\right). \end{array}$$
This leads to the following κ-deformation of the Bloch equation:(27)$$\begin{array}{c} -\frac{\partial }{\partial \beta }{\hat{\rho }}_{\kappa }\left(\beta \right)=\sum_{i}E_{i}\frac{\exp_{\kappa }\left(-\beta E_{i}\right)}{u_{\kappa }\left[(\exp_{\kappa }\left(-\beta E_{i}\right)\right]}\;|\psi_{i}\rangle \langle \psi_{i}|=\frac{\hat{H}}{u_{\kappa }\left[\exp_{\kappa }\left(-\beta \hat{H}\right)\right]}\;{\hat{\rho }}_{\kappa }\left(\beta \right). \end{array}$$
Again, by changing the parameter β to t=−lnβ and using the relation (Equation 52), we have:(28)$$\begin{array}{c} \frac{d}{dt}{\hat{\rho }}_{\kappa }\left(t\right)=-\frac{\ln_{\kappa }\left[{\hat{\rho }}_{\kappa }\left(t\right)\right]}{u_{\kappa }\left[{\hat{\rho }}_{\kappa }\left(t\right)\right]}\;{\hat{\rho }}_{\kappa }\left(t\right), \end{array}$$
which can be regarded as a κ-deformation of the Gompertz rule.

Differentiating (Equation 27), again with respect to β, we obtain the following nonlinear differential equation:(29)$$\begin{array}{c} \left(1+\kappa^{2}\beta^{2}{\hat{H}}^{2}\right)\frac{\partial^{2}{\hat{\rho }}_{\kappa }\left(\beta \right)}{\partial \beta^{2}}+\kappa^{2}\beta {\hat{H}}^{2}\frac{\partial {\hat{\rho }}_{\kappa }\left(\beta \right)}{\partial \beta }-{\hat{H}}^{2}\;{\hat{\rho }}_{\kappa }\left(\beta \right)=0. \end{array}$$
This differential equation reminds us of the research work [24] on the quantum free particle on the two-dimensional hyperbolic plane. The relevant two-dimensional Schrödinger equation was separable in the κ-dependent coordinate system $\left(z_{x},y\right)$ with $z_{x}:=x/\sqrt{1+\kappa^{2}y^{2}}$. The Schrödinger equation ${\hat{H}}_{1}\Psi =e_{1}\Psi$ for the first partial Hamiltonian ${\hat{H}}_{1}$ leads to the following differential equation with the variable $z_{x}$ alone:(30)$$\begin{array}{c} \left(1+\kappa^{2}z_{x}^{2}\right)\frac{d^{2}\Psi \left(z_{x}\right)}{dz_{x}^{2}}+\kappa^{2}z_{x}\frac{d\Psi (z_{x})}{dz_{x}}+\mu \Psi \left(z_{x}\right)=0,\;\mu :=\frac{2m}{\hbar^{2}}e_{1}. \end{array}$$
In the limit of κ→0, this differential equation reduces to the standard time-independent Schrödinger equation: $d^{2}\Psi \left(x\right)/dx^{2}+\mu \Psi \left(x\right)=0$. Cariñena et al. [24] obtained the solution of the differential Equation (Equation 30) as the κ-deformed plane wave (in our notations):(31)$$\begin{array}{c} \Psi \left(z_{x}\right)=\exp\left[\pm i\;\frac{\mu }{\kappa }\mathrm{arsinh}\left(\kappa \;z_{x}\right)\right], \end{array}$$
which is regarded as an arsinh-type deformation.

## 4. The κ-Addition and the Law of Large Number

Next, we considered the κ-addition from the viewpoint of the law of large numbers (LLN), which plays a central role in probability, statistics, and statistical physics [25]. The κ-addition [4] is defined by:(32)$$\begin{array}{c} x\overset{\kappa }{⊕}y:=x\sqrt{1+\kappa^{2}y^{2}}+y\sqrt{1+\kappa^{2}x^{2}}. \end{array}$$
This deformation of the additive rule comes from the addition rule of the inverse hyperbolic sine function as follows. For a,b∈R, the addition rule is written as:(33)$$\begin{array}{c} \mathrm{arsinh}\left(a\right)+\mathrm{arsinh}\left(b\right)=\mathrm{arsinh}\left(a\sqrt{1+b^{2}}+b\sqrt{1+a^{2}}\right). \end{array}$$
By setting a=κx and b=κy, we obtain:(34)$$\begin{array}{cc} \mathrm{arsinh}(\kappa x) & +\mathrm{arsinh}\left(\kappa y\right)=\mathrm{arsinh}\left(\kappa x\sqrt{1+\kappa^{2}y^{2}}+\kappa y\sqrt{1+\kappa^{2}x^{2}}\right)\\ \; & =\mathrm{arsinh}\left[\kappa (x\overset{\kappa }{⊕}y)\right]. \end{array}$$
This relation is equivalent to the definition (Equation 32). The additive relation (Equation 34) is readily generalized to:(35)$$\begin{array}{c} \sum_{i=1}^{n}\mathrm{arsinh}\left(\kappa x_{i}\right)=\mathrm{arsinh}\left[\kappa (x_{1}\overset{\kappa }{⊕}x_{2}\overset{\kappa }{⊕}\cdots \overset{\kappa }{⊕}x_{n})\right]. \end{array}$$
By applying this relation to the Boltzmann factor $\exp\left[-\beta \sum_{i=1}^{n}K_{\kappa \beta }\left(p_{i}\right)\right]$ with respect to the κ-deformed kinetic energy [8] with m=1:(36)$$\begin{array}{c} \sum_{i=1}^{n}K_{\kappa \beta }\left(p_{i}\right):=\sum_{i=1}^{n}\frac{1}{\kappa \beta }\mathrm{arsinh}\left(\kappa \beta \frac{p_{i}^{2}}{2}\right), \end{array}$$
we have:(37)$$\begin{array}{cc} \exp\left[-\beta \sum_{i=1}^{n}K_{\kappa \beta }\left(p_{i}\right)\right] & =\exp\left[-\frac{1}{\kappa }\mathrm{arsinh}\left\{\kappa \left(\beta \frac{p_{1}^{2}}{2}\overset{\kappa }{⊕}\beta \frac{p_{2}^{2}}{2}\overset{\kappa }{⊕}\ldots \overset{\kappa }{⊕}\beta \frac{p_{n}^{2}}{2}\right)\right\}\right]\\ \; & =\exp_{\kappa }\left[\left(-\beta \frac{p_{1}^{2}}{2}\right)\overset{\kappa }{⊕}\left(-\beta \frac{p_{2}^{2}}{2}\right)\overset{\kappa }{⊕}\ldots \overset{\kappa }{⊕}\left(-\beta \frac{p_{n}^{2}}{2}\right)\right]\\ \; & =\exp_{\kappa }\left[-\beta \frac{p_{1}^{2}}{2}\right]\exp_{\kappa }\left[-\beta \frac{p_{2}^{2}}{2}\right]\ldots \exp_{\kappa }\left[-\beta \frac{p_{n}^{2}}{2}\right]=\prod_{i=1}^{n}\exp_{\kappa }\left[-\beta \frac{p_{i}^{2}}{2}\right]. \end{array}$$
Note that the κ-exponential of the κ-summation of each term $-\beta \frac{p_{i}^{2}}{2}$ in the second line is expressed as a factorized form in the last line.

It is well known that LLN plays a fundamental role in statistical physics [25]. apiński [26] showed that the standard LLN yielded the most probable state of the system, which equaled the point of maximum of the entropy and this point could be either Maxwell–Boltzmann statistics or Bose–Einstein statistics, or Zipf–Mandelbort law. McKeague [27] studied the central limit theorems under the special theory of relativity based on the κ-additivity. Scarfone [28] studied the κ-deformation of Fourier transform and discussed the limiting distribution of the κ-sum of statistically independent variables. The κ-additivity extension of the strong LLN was shown in [27] and it stated that if $X_{i}$ were iid with finite mean, then:(38)$$\begin{array}{c} \frac{X_{1}}{n}\overset{\kappa }{⊕}\frac{X_{2}}{n}\overset{\kappa }{⊕}\ldots \overset{\kappa }{⊕}\frac{X_{n}}{n}\;\to \;\frac{1}{\kappa }\mathrm{arsinh}\left[\kappa \langle X\rangle \right]{}_{a.s.}, \end{array}$$
where a.s. stands for almost surely, i.e., the above sequence of the random variables $X_{i}$ converges almost surely, and $\langle X\rangle$ is the standard average of the random variable *X*. Of course, in the limit of κ→0, the relation (Equation 38) reduced to the standard strong LLN. Note that the converged value in (Equation 38) was the arsinh-type deformation of the average $\langle X\rangle$. In this way, the κ-additivity extension of the strong LLN supports the arsinh-type deformation of the average of a stochastic variable *X*.

## 5. Contact Density Dynamics

Nosé-Hoover (NH) thermostat [29,30] is a famous deterministic algorithm for constant-temperature molecular dynamics simulations. Based on the idea of NH thermostat, several improved versions were proposed. Among them, contact density dynamics (CDD) [31] is an algorithm based on contact Hamiltonian systems and generates any prescribed target distribution in physical phase space. The dynamical equations of CDD are the following:
{(39a)dqidt=∂h(q,p,S)∂pi,(39b)dpidt=−∂h(q,p,S)∂qi+∂h(p,q,S)∂Spi,(39c)dSdt=−pi∂h(q,p,S)∂pi+h(q,p,S),
where *S* is the thermostatting variable, $q_{i}$ and $p_{i}$ are the *i*-th component (i=1,2,…,n) of *n*-dimensional vectors, respectively. Here h(q,p,S) denotes the contact Hamiltonian which is formed as:(40)$$\begin{array}{c} h\left(q,p,S\right)={\left(\rho_{t}\left(q,p\right)f\left(S\right)\right)}^{-\frac{1}{n+1}}, \end{array}$$
with a target distribution $\rho_{t}\left(q,p\right)$ on 2n-dimensional Γ-space and a normalized distribution f(S) for the thermostatting variable *S*. As in the case of Reference [29,30], we also chose f(S) as the logistic distribution with scale 1 and mean c=0.0:(41)$$\begin{array}{c} f\left(S\right)=\frac{\exp(S-c)}{{\left(1+\exp\left(S-c\right)\right)}^{2}}. \end{array}$$

Utilizing this CDD algorithm, the κ-deformed exponential distributions were simulated. The target distribution $\rho_{t}\left(q,p\right)$ was the one-dimensional (n=1) κ-deformed Gaussian function:(42)$$\begin{array}{cc} \rho_{t}\left(q,p\right) & =\frac{1}{Z_{\kappa }\left(\beta \right)}\exp\left[-\beta H_{\kappa }\left(q,p\right)\right]=\frac{1}{Z_{\kappa }\left(\beta \right)}\exp\left[-\frac{1}{\kappa }\mathrm{arsinh}\left(\kappa \beta \frac{p^{2}}{2}\right)\right]\exp\left[-\beta \frac{q^{2}}{2}\right], \end{array}$$
where the associated Hamiltonian was:(43)$$\begin{array}{cc} H_{\kappa }\left(q,p\right) & =\frac{1}{\kappa \beta }\mathrm{arsinh}\left(\kappa \beta \frac{p^{2}}{2}\right)+\frac{q^{2}}{2}, \end{array}$$
and the normalization factor $Z_{\kappa }\left(\beta \right)$ [4] was:(44)$$\begin{array}{c} Z_{\kappa }\left(\beta \right)=\frac{\pi }{\beta }\frac{\sqrt{\frac{2}{\kappa }}\Gamma \left(\frac{1}{2\kappa }-\frac{1}{4}\right)}{\left(\frac{\kappa }{2}+1\right)\Gamma \left(\frac{1}{4}+\frac{1}{2\kappa }\right)}. \end{array}$$

In general, the kinetic energy can be defined by:(45)$$\begin{array}{c} K\left(p\right):=\int_{0}^{p}v\left(p\right)dp, \end{array}$$
where v(p) denotes the constitutive relation between the velocity *v* and the canonical momentum *p*. In the standard case of v(p)=p/m with m=1, we have $K\left(p\right)=p^{2}/2$. In the case of the Hamiltonian (43), from (39a) we have:(46)$$\begin{array}{cc} v_{\kappa }\left(p\right):=\frac{dq}{dt}=\frac{\partial H_{\kappa }\left(q,p\right)}{\partial p} & =\frac{p}{u_{\kappa }\left[\exp_{\kappa }\left(-\beta \frac{p^{2}}{2}\right)\right]}=\frac{p}{\sqrt{1+\kappa^{2}{\left(\beta \frac{p^{2}}{2}\right)}^{2}}}. \end{array}$$
It is worthwhile to note that the $v_{\kappa }\left(p\right)$ had a β (or temperature) dependency when κ≠0. Then the corresponding kinetic energy $K_{\kappa }\left(p\right)$ was the first term $\frac{1}{\kappa \beta }\mathrm{arsinh}\left(\kappa \beta \frac{p^{2}}{2}\right)$ in (43), which could be regarded as a κ-deformation of the standard kinetic energy $p^{2}/2$.

We performed a number of CDD simulations for the target state (42) with different parameters and initial conditions. As an example, Figure 1 shows the phase space orbit and the histogram of the frequencies of the momentum *p* for a typical result of the CDD simulation of the target state (42) with β=0.2, κ=0.4. The initial conditions used are also denoted in the figure captions.

The CDD simulated result obeys ergodicity, as can be seen from the well distributed points in the phase space in Figure 1a. Note that the momentum distribution in the histogram of Figure 1b was well fitted with the κ–Gaussian distribution, which was cased by the arsinh-type deformation of the kinetic energy $p^{2}/2$.

Note also that for the κ-deformed Hamiltonian (42), we have [8]:(47)$$\begin{array}{c} \langle p\frac{\partial }{\partial p}H_{\kappa }\left(q,p\right)\rangle =\frac{1}{\beta } \end{array}$$
which reminds us of *a generalization of equipartition theorem* [32]: $\langle p\frac{\partial }{\partial p}H\rangle =k_{B}T$, where H is the Hamiltonian of a system in thermal equilibrium with the temperature *T*.

## 6. Conclusions

We considered the κ-deformations of some quantities concerning statistical physics and pointed out some unexpected relations among different fields, such as statistical mechanics, mathematical biology and evolutional game theory. We especially focused on the arsinh-type deformation of the ratio $\beta p^{2}/2$ of kinetic energy to the average thermal energy $k_{B}T=1/\beta$. With the help of the thermostat (CDD) algorithm we performed the relevant numerical simulations for the Hamiltonian with the arsinh-type deformation of kinetic energy term and showed the resultant momentum distribution was the κ–Gaussian distribution.

Finally, we would like to point out a relation which might be suggested for future research. Let us consider the κ-deformed energy density of state $\Omega_{\kappa }\left(U\right)$:(48)$$\begin{array}{c} \Omega_{\kappa }\left(U\right):=\exp_{\kappa }\left(\frac{U}{k_{B}T_{c}}\right)=\exp\left[\frac{1}{\kappa }\mathrm{arsinh}\left(\kappa \frac{U}{k_{B}T_{c}}\right)\right], \end{array}$$
which is the κ-deformation of the energy density of state $\exp(U/k_{B}T_{c})$ for the thermal reservoir with a constant-temperature $T_{c}$ (Boltzmann reservoir [33]). In other words, $\ln\Omega_{\kappa }\left(U\right)$ is regarded as the arsinh-type deformation of the ratio $U/(k_{B}T_{c})$. The Boltzmann temperature T(U) for this κ-deformed thermal reservoir is given by:(49)$$\begin{array}{c} \frac{1}{k_{B}T\left(U\right)}:=\frac{d\ln\Omega (U)}{dU}=\frac{\frac{1}{k_{B}T_{c}}}{\sqrt{1+\kappa^{2}{\left(\frac{U}{k_{B}T_{c}}\right)}^{2}}}. \end{array}$$
Rearranging this relation leads to:(49)$$\begin{array}{cc} k_{B}T\left(U\right) & =\sqrt{{\left(\kappa U\right)}^{2}+{\left(k_{B}T_{c}\right)}^{2}}, \end{array}$$
which reminds us of the relativistic energy–momentum relation: $E\left(p\right)=\sqrt{{\left(cp\right)}^{2}+{\left(mc^{2}\right)}^{2}}$.

## Figures and Tables

**Figure 1 entropy-25-00292-f001:**
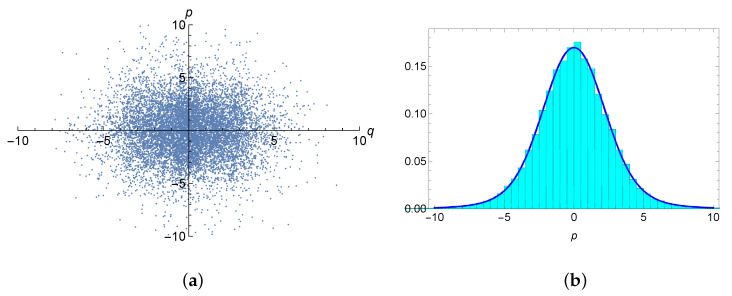
The simulated results of the CDD simulations of the target distribution (Equation 41) with κ=0.4 and β=0.2. (**a**) the phase (*q*-*p*) space orbit of the κ-deformed distribution. The $1.5\times 10^{4}$ points of a simulated orbit with the initial condition ($q_{0}=0.1,p_{0}=0.1$, and $S_{0}=0.9$ are shown. (**b**) the histogram of the frequencies for *p* and the corresponding momentum κ-distribution (blue solid curve).

## Data Availability

Not applicable.

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
