# Peer review of "On the Kaniadakis Distributions Applied in Statistical Physics and Natural Sciences"

_entropy, 2023, doi:10.3390/e25020292_

Round 1

Reviewer 1 Report

In this manuscript, it is considered the arsinh-type deformations for constructive relations in the field of natural sciences and statistical physics. Generally, any linear constitutive relation describes an idealized solution, but non-linearity may be considered to describe more realistic physical systems. Authors exemplify this situation using k-deformed exponential function and k-deformed logarithmic function in the generalization of Hooke’s law and Ohm’s law, which recovers the original relation when k tends to zero. In this manuscript, authors focus on k-deformed arsinh to represent physical quantities.

Twenty years ago, Giorgio Kaniadakis presented a new one-parameter deformation for the exponential function and others deformed functions, which describe power-law asymptotic behavior. He did this to obtain the novel Kaniadakis deformed statistic. Since that, such statistics and deformed functions have been applied in several works of many fields of applications. This manuscript is part of this research line. It is original and represents a significant contribution.

As an application of the formalism presented in the manuscript, authors studied the numerical simulations of the thermostat algorithm for the Hamiltonian with the deformed kinetic energy based in the k-deformed arsinh function, showing that the momentum distribution is the k-Gaussian distribution.

I miss a justification for the authors' decision to use k=0.4 in the results section or if it was an arbitrary decision. If so, why did they only plot in Figure 1 for that value?

The mathematical formalism is adequately described, and the numerical results are clearly presented in the manuscript. The results satisfactorily support the conclusions of the manuscript.

The manuscript needs minor editing.

I recommend the publication of this manuscript in Entropy journal.

Author Response

We thank the reviewer for valuable opinions.

As the reviewer pointed out, the selected value of \kappa=0.4 has a no special meaning. In order to avoid such misleading, we added "As an example" before

the sentence in line 197.

Thank you for pointing out this.

Reviewer 2 Report

This is a well-written survey article that covers material that should be of interest to a wide audience of physicists, mathematical biologists and statisticians.

The English needs be checked carefully in places: for example, on p.6, line 139 "as a arsinh-type deformation" should be "as an arsinh-type deformation"

I also noticed a typo in the title of one of the references.  The reference [22] is in fact

McKeague. I.W.  Central limit theorems under special relativity ...

and not "Central limit theorem under special relativity."

Author Response

Thank you for your review.

We corrected the mistype which you pointed out.

Reviewer 3 Report

The paper reviews a number of known results about Kaniadakis' deformed entropy and adds new simulation results. The claim is made that the formalism is in particular relevant for constitutive relations.

The paper contains some inaccuracies that must be improved.

Equation (13) is a generalized LV equation only if one assumes that the functions f_i are affine. Otherwise, it is nothing but an example of a first order ODE. In particular, (16) is not a generalized LV equation.

Essential for statistics and statistical physics is that the density function/density matrix is normalized. If one adds the normalization then the links discussed in the paper disappear. In other words, the relevance for statistics and statistical physics is nihil.
For instance:
l119: "In order to determine the canonical density matrix, ..."
The canonical density matrix of statistical physics has trace 1.
Hence, the claim that (22) is a canonical density matrix is misleading, even deceiving.

The Bloch equation is of the form dM/dt=gamma M x B - g M.
I do not see the link with (23).

Why do you refer to the unpublished work [14] of M Harper and not one of his published papers?

Lapinski is not the first to prove that LLN yields the most probable state.
This was a hot topic around 1970. John Lewis from Dublin was one of the leading authors on this topic.

There are to many typing errors / language errors. For instance:

Keywords: 'Voltela' --> 'Volterra'

p2, l46: period missing at the end of the sentence.
p3,l70: 'realize' --> 'realized'
p3,l81: 'concerning on statistical physics.' --> 'concerning statistical physics.'
p3, l84: '...  is pointed out.' --> '... are pointed out.'
p4, l92: 'we e’d' --> 'we 'd' or 'we would'
p4, l92: '... RE' --> '... an RE' or '... a replicator equation'
p4, l92: 'related with' --> 'related to'
p4, l98: 'tables mortality' --> 'mortality tables' or 'tables of mortality'
p7, l171: ' followings' --> 'the following'.
p8, l188: 'We have performed several numbers ...' --> 'We have performed a large number ...'

...

Author Response

Thank you for your review.

We reply to your all comments in the attached pdf file "Rep2Refree.pdf". 

Reviewer 4 Report

In the submitted manuscript, the authors have considered some quantities relevant for statistical physics and the behaviour of systems described by generalized Lotka-Volterra evolution, in the framework of the κ-deformation of Kaniadakis entropy. In particular, the authors are pivoting their work about the arsinh-type deformation. The authors have also provided related numerical simulations.

The submitted work consists in a remarkably clear and well-written mathematical exposition of their approach. The rigour of the applied method allowed the authors to propose some new hints that can be fruitful for further researches. Before publication, I would like to suggest some minimal changes. In the case of the Gompertz function, please add some references to applications of it. The same please for Lotka-Volterra, in particular for game theories. Please revise the typos in keywords: Lotka-Voltela in Lotka-Volterra.

Author Response

Thank you for your review.

We added some references for Gompertz function and Lotka-Volterra equation.

Also we corrected the mistype which you pointed out.

Round 2

Reviewer 3 Report

Equation (13) still is strange because almost any set of partial differential equations can be written in this form because the prefactors y_i can be included in the function f_i.

Author Response

As the reviewer pointed out, if the pre-factor y_i is included into the function f(y), any ODE would be written in the form  dy_i /dt = f(y). However we never do such a way.  At least, in some literature[14,15, 19], Eq.(13) is named GLV equation.Furthermore, what is more important is not the name of Eq. (13) but the fact that the solutions y_i in Eq. (13) are related to the solutions x_i of the replicator equation (RE) as explained in Appendix B.